# Robust and Lightweight Deep Learning Model for Industrial Fault Diagnosis in Low-Quality and Noisy Data

Jaegwang Shin and Suan Lee *

School of Computer Science, Semyung University, Jecheon 27136, Republic of Korea
* Correspondence: suanlee@semyung.ac.kr; Tel.: +82-43-649-1273

**Abstract:** Machines in factories are typically operated 24 h a day to support production, which may result in malfunctions. Such mechanical malfunctions may disrupt factory output, resulting in financial losses or human casualties. Therefore, we investigate a deep learning model that can detect abnormalities in machines based on the operating noise. Various data preprocessing methods, including the discrete wavelet transform, the Hilbert transform, and short-time Fourier transform, were applied to extract characteristics from machine-operating noises. To create a model that can be used in factories, the environment of real factories was simulated by introducing noise and quality degradation to the sound dataset for Malfunctioning Industrial Machine Investigation and Inspection (MIMII). Thus, we proposed a lightweight model that runs reliably even in noisy and low-quality sound data environments, such as a real factory. We propose a Convolutional Neural Network–Long Short-Term Memory (CNN–LSTM) model using Short-Time Fourier Transforms (STFTs), and the proposed model can be very effective in terms of application because it is a lightweight model that requires only about 6.6% of the number of parameters used in the underlying CNN, and has only a performance difference within 0.5%.

**Keywords:** fault diagnosis; deep learning; CNN; image representation; feature extraction

## 1. Introduction

Advancements in industrial technology have led to factories becoming increasingly more automated, predictive maintenance technologies evolving considerably, and automated smart factories being developed for efficient large-scale manufacturing [1]. However, as manufacturing equipment is typically in constant operation, there is a high probability of mechanical malfunction that may disrupt factory output and result in financial losses or human casualties. These severe failures are typically triggered by minor failures. Thus, the early detection of minor failures can considerably mitigate any harm, and tools for fault detection are being developed [2–4]. However, monitoring the machines for 24 h a day is cumbersome and detecting the minor failures visually is difficult. Therefore, in this study, we investigated two deep learning models for detecting failures by monitoring the sound generated by machines during operation [5,6].

This study used the MIMII dataset [7] that contains recordings of sounds that may be generated in a factory. The recordings comprise sounds generated by four machines: a fan, a pump, a valve, and a sliding rail. We used discrete wavelet transforms (DWT), Hilbert transforms, and short-time Fourier transforms (STFTs) to convert features of sound data into images after extraction and use them as inputs to deep learning models. Furthermore, we investigated model performance under three recording qualities (16 K, 8 K and 4 K) and conditions with different signal-to-noise ratios (SNRs) (−6 dB, 0 dB, 6 dB). This was performed in order to take into account the real factory environment and the existence of ambient factory noise.

We have created a CNN-based model with a structure similar to that of existing studies, and propose a lightweight CNN–LSTM-based model available in real-world industries. Our

proposed model has shown that it is robust with excellent results even on low-quality and noisy sound data. Compared to the underlying CNN model, our proposed CNN–LSTM model showed superior performance, even with only 6.6% of the number of parameters. The contributions of this study are as follows:

- We investigated the impact of three sound data pre-processing methods (the Hilbert transform, DWT, and the STFT) on deep learning model performance.
- We propose a CNN–LSTM lightweight model with the STFT, and show an efficient performance even with a very small number of parameters. Existing studies have been conducted using models with transfer learning from pre-trained models or models with many parameters [8–11].
- We investigated the model performance on lower data qualities and with random noise, with the goal of gauging model performance in real-world factory settings.

This paper is organized as follows. Section 2 discusses previous research on anomaly detection in manufacturing. Section 3 presents the feature engineering techniques for the sound of equipment utilized in this study. Section 4 describes the architecture of the model proposed in this study. Section 5 describes the dataset utilized in the experiment, the environment, and the results of various experiments for each model. Section 6 discusses the proposed model by comparing it with existing similar models. Finally, Section 7 discusses conclusions and future work.

## 2. Related Works

Machine failure on the assembly floor can halt production, and repairing such machines incurs considerable cost. Therefore, numerous studies have been conducted to obtain industry data. The MIMII dataset [7] is widely used as representative data in many studies. For example, the MIMII dataset has been used in deep learning models, such as in a study to detect outliers using SCRLSTM, which is an ensemble model combining CNN and LSTM [8], and an ensemble model based on the EfficientNet model for outlier detection. Furthermore, the dataset has been previously used in a detection model [9], a study comparing CNN–LSTM and CNN–GRU [10], and a study of a model robust to noise [11]. In this paper, we also propose a CNN-based model and a combination of CNN and LSTM.

Studies using semi-supervised, unsupervised, and self-supervised learning methods include outlier detection, using self-supervised complex networks [12], outlier detection, using an unsupervised domain adaptation method [13], a study on a learning MIMII dataset through the semi-supervised learning of the RawdNet model [14], and an outlier detection study using self-supervised learning that utilizes a contrast learning framework [15]. MIMII dataset preprocessing has been realized by removing noise from MIMII datasets, via NMF and nnCP models [16]. Outlier detection was performed by improving the performance of the DCNN–LSTM, using the Hilbert transform [17].

Studies have been conducted on preprocessing the data to improve the performance of the anomaly detection model. For example, the STFT and DWT were used in a study on industrial equipment frequency analysis [18] and a study investigated the performance of the CatGAN model using the STFT [19]. Furthermore, the Hilbert transform was used for studying ECG data [20] and mechanical vibration [21]. A study used DWT in performing audio analysis [22]. Many industries and applications use technologies such as DWT, the Hilbert transformation, and the STFT for efficient signal processing. Our model also uses these techniques for feature extraction to classify sounds for fault diagnosis.

Many studies have been conducted on model performance, model learning methodologies, and data preprocessing. However, few studies have considered the real environment. A factory may have considerable ambient noise, or the factory environment may not be suitable for capturing data of decent quality. Thus, we investigated models that exhibit robust performance, even when noise is introduced, or data quality is degraded.

## 3. Feature Engineering

In this paper, feature extraction and transformation processes were performed to use sound datasets as input into the deep learning model for fault diagnosis, as shown in Figure 1. First, diagnostic equipment, used in the environment of an industrial field, may have limited memory and bandwidth. Therefore, we experimented by lowering the basic quality of 16 K to 8 K and 4 K in the sound dataset. In addition, a comparative experiment was conducted using SNR values of −6 dB, 0 dB, and 6 dB to consider the noise environment. Second, features were extracted from sound using the DWT, Hilbert, and the STFT, and converted into images. Finally, the image created through the processing process was used as an input to the deep learning model.

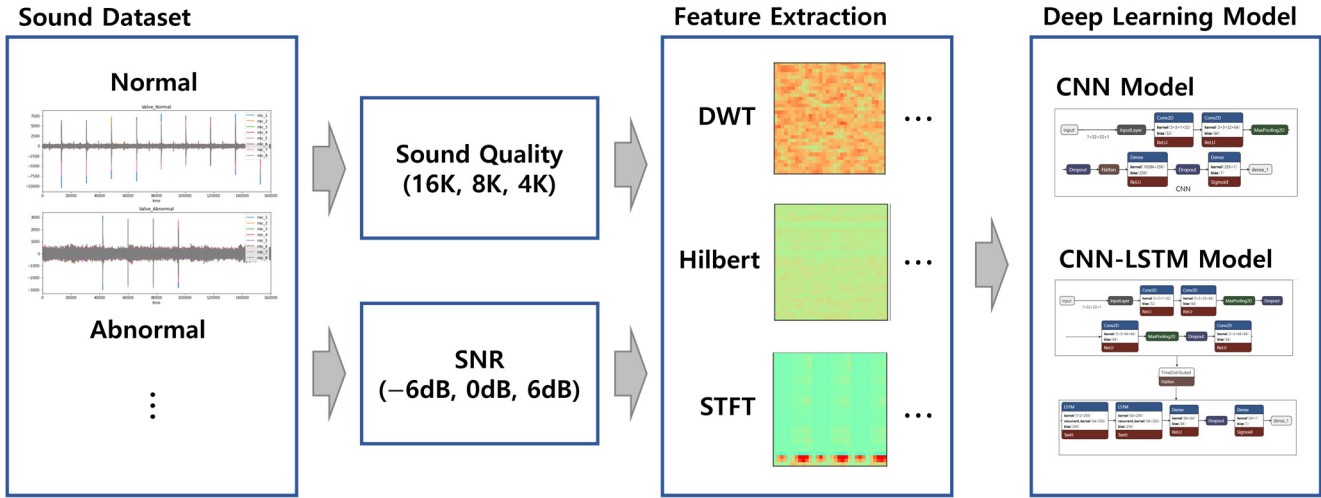

**Figure 1.** Step-by-step processing of sound data for deep learning models.

### 3.1. Discrete Wavelet Transform (DWT)

DWT is a wavelet-based signal decomposition algorithm. A high-pass filter is used to produce coefficients in DWT and a low-pass filter is used to obtain approximation coefficients. We used a low-pass filter to obtain approximation coefficients, and used db4 as the wavelet. Following the application of DWT to the sound data, we generated a 32 × 32 image from it to accommodate the deep learning model. Figures 2–4 display the images generated from five random slide rail sounds, transformed using DWT, at quality levels 4 K, 8 K, and 16 K. Ground truth is the actual result and Predicted is the model's prediction. The fan, valve, pump, and slide rail data of the MIMII dataset were transformed into images.

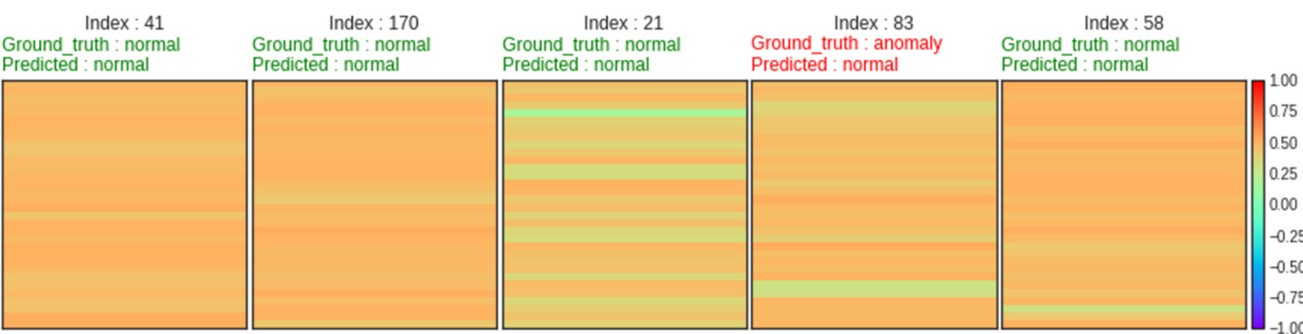

**Figure 2.** Slide rail data (4 K) DWT image encoding.

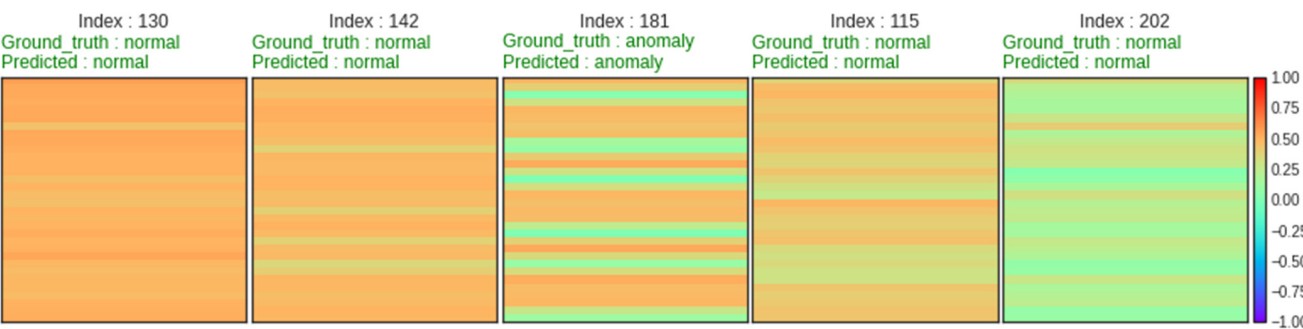

**Figure 3.** Slide rail data (8 K) DWT image encoding.

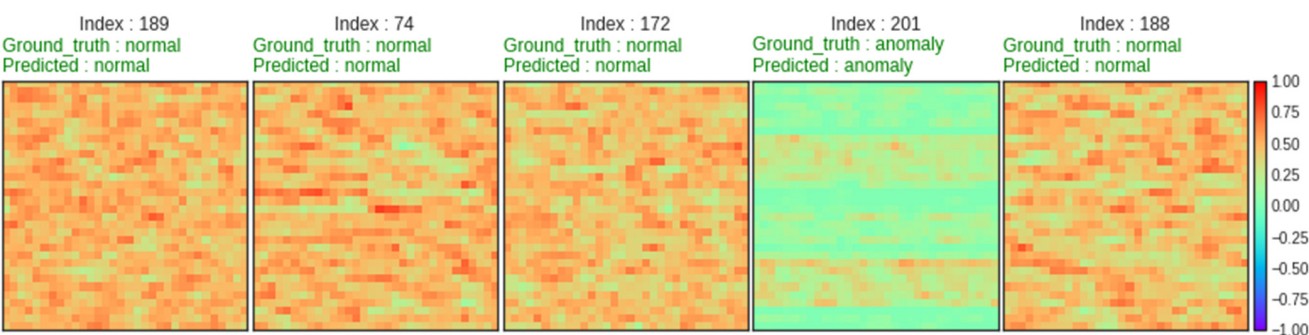

**Figure 4.** Slide rail data (16 K) DWT image encoding.

### 3.2. Hilbert Transform

The Hilbert transform is a linear operator used in signal processing that only modifies the phase and not the frequency. Positive frequencies are phase shifted by $-\pi/2$, and negative frequencies are phase shifted by $\pi/2$. Thus, similar to how a sine signal is produced when a cosine signal is shifted by $-\pi/2$, the Hilbert transform outputs a negative frequency when given a positive one, and outputs a positive frequency when given a negative one. A negative frequency was obtained in this study because the absolute value of a positive frequency was utilized. The Hilbert transform expands real signals into the complex number dimension. Therefore, the model can acquire more information to distinguish between normal and aberrant data in the MIMII dataset, resulting in enhanced performance. Our model was trained on $32 \times 32$ images obtained from the MIMII dataset, using the Hilbert transform.

Figures 5–7 display the results of this transformation, applied to slide rail data in the same manner as that of the DWT. Similarly, Ground_truth is the actual result and Predicted is the model's prediction.

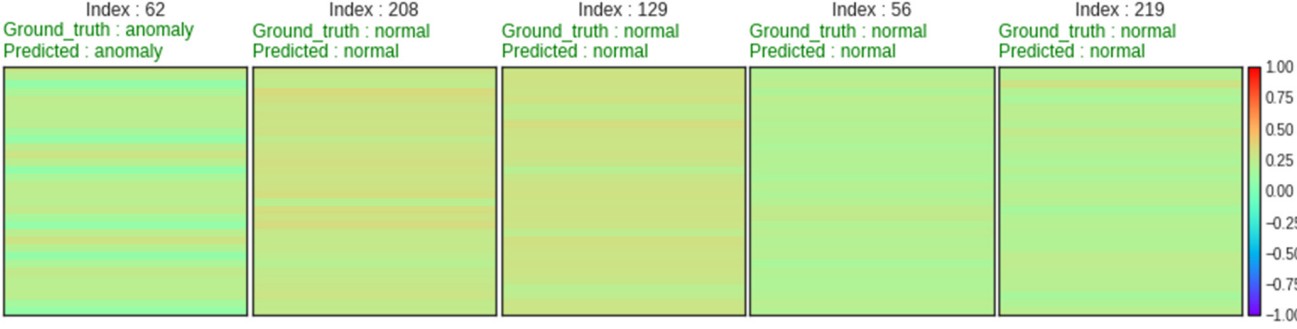

**Figure 5.** Slide rail data (4 K) Hilbert transform image encoding.

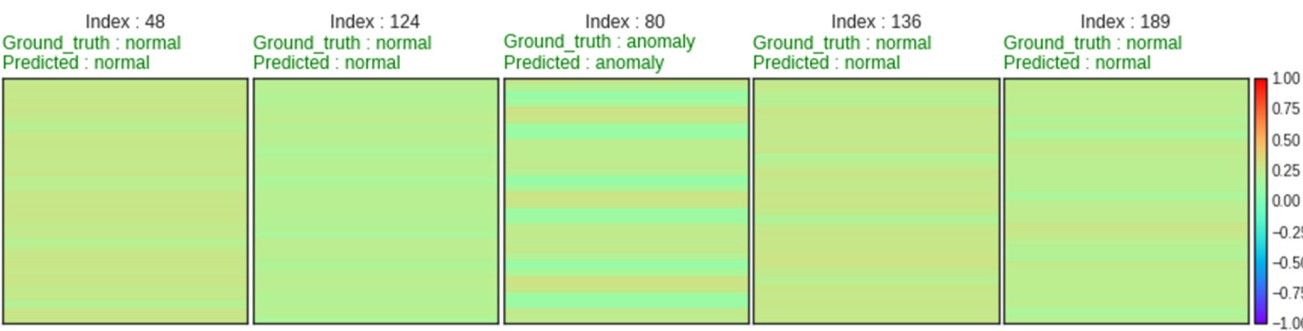

**Figure 6.** Slide rail data (8 K) Hilbert transform image encoding.

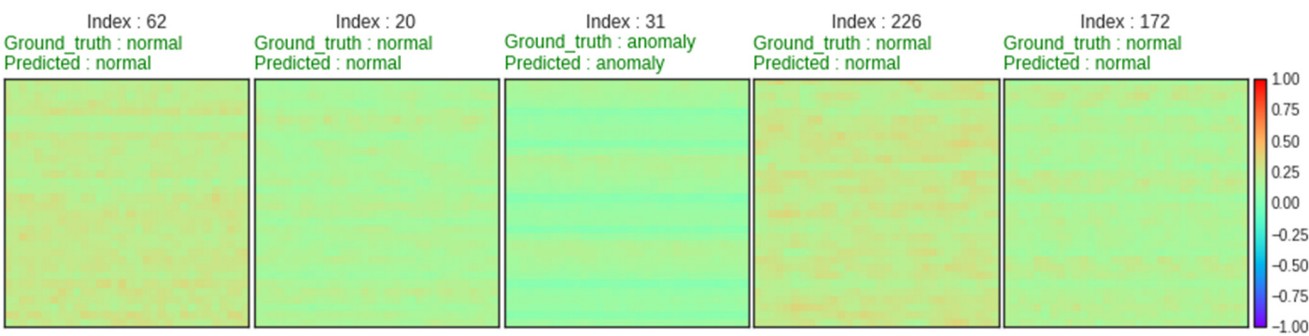

**Figure 7.** Slide rail data (16 K) Hilbert transform image encoding.

*3.3. Short-Time Fourier Transform (STFT)*

The STFT is used for separating data into small segments based on the Fourier transform and the enduring Fourier transform. The STFT is similar to the FFT, but it simplifies time series data, can determine frequencies that occur in intervals and expands the dimension in a manner comparable to that of the Hilbert transform. Thus, a one-dimensional signal is converted to a two-dimensional signal, enabling the model to obtain additional information. The STFT is mainly used by converting into spectrogram images, and we also used MIMII datasets by converting them into spectrogram images. We reduced it to a $32 \times 32$ size image for use as an input to a deep learning model. The results of translating slide rail data into images, using the STFT, are displayed in Figures 8–10. Similarly, Ground_truth is the actual result, whereas Predicted is the prediction result of the model. Looking at the results, there is a difference between the figure of normal and anomaly.

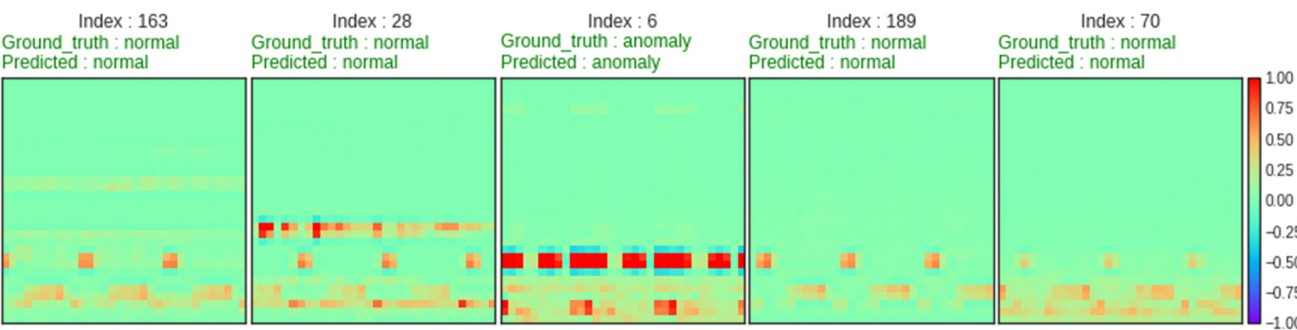

**Figure 8.** Slide rail data (4 K) STFT image encoding.

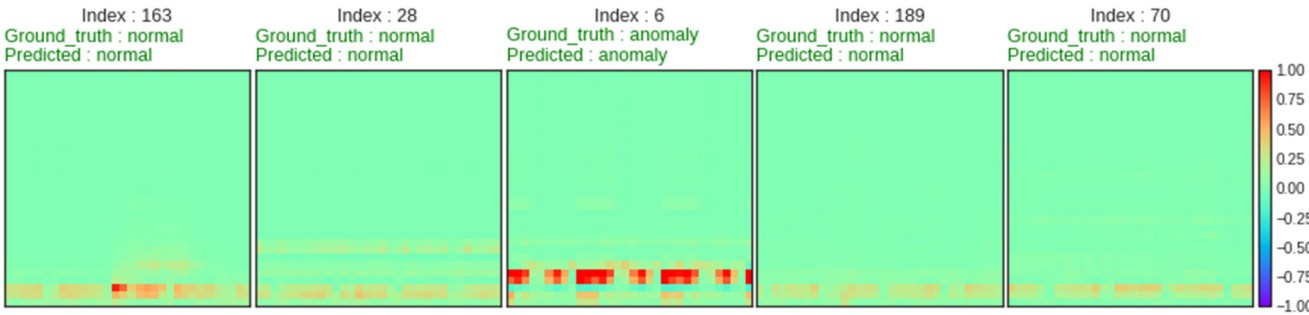

**Figure 9.** Slide rail data (8 K) STFT image encoding.

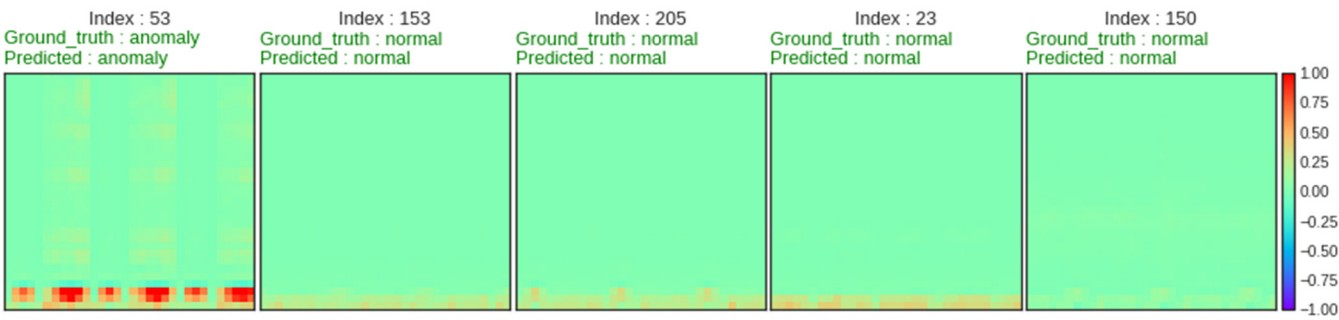

**Figure 10.** Slide rail data (16 K) STFT image encoding.

## 4. Model Architecture

### 4.1. CNN

The architecture of the CNN-based model is displayed in Figure 11. The model was configured to receive images of size $32 \times 32$ as input. The images were passed through two Conv2D layers. The activation function was the sigmoid function, and the loss function was binary_crossentropy. The Adam optimizer was used to adjust the learning rate to 0.001. Early stopping was used to stop training when the validation loss was minimal. The total number of parameters was 4,213,633.

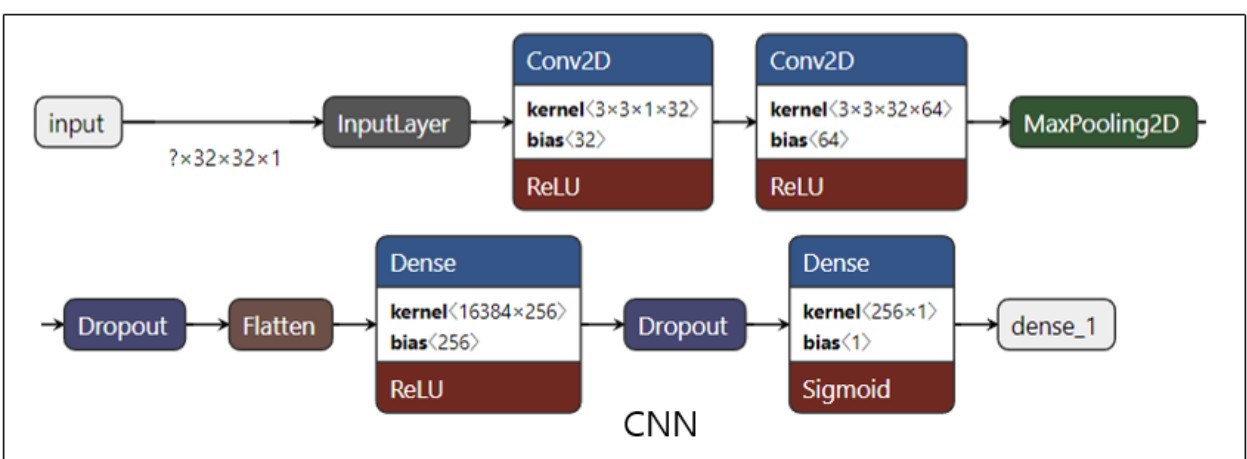

**Figure 11.** CNN model architecture.

### 4.2. CNN–LSTM

The CNN–LSTM architecture is displayed in Figure 12. Similar to the CNN-based model, the input size was $32 \times 32$, the activation function was the sigmoid function, and binary_crossentropy was the loss function. However, in this model, the input passed through four Conv2D layers and two LSTM layers. To achieve the same requirements as

the CNN, RMSprop was used to set the learning rate to 0.001. Early stopping was also used to stop learning when the validation loss was minimal. The CNN–LSTM only required 277,633 parameters, considerably less than the CNN. Thus, the CNN–LSTM model should be easier to implement in an embedded system or AIoT equipment.

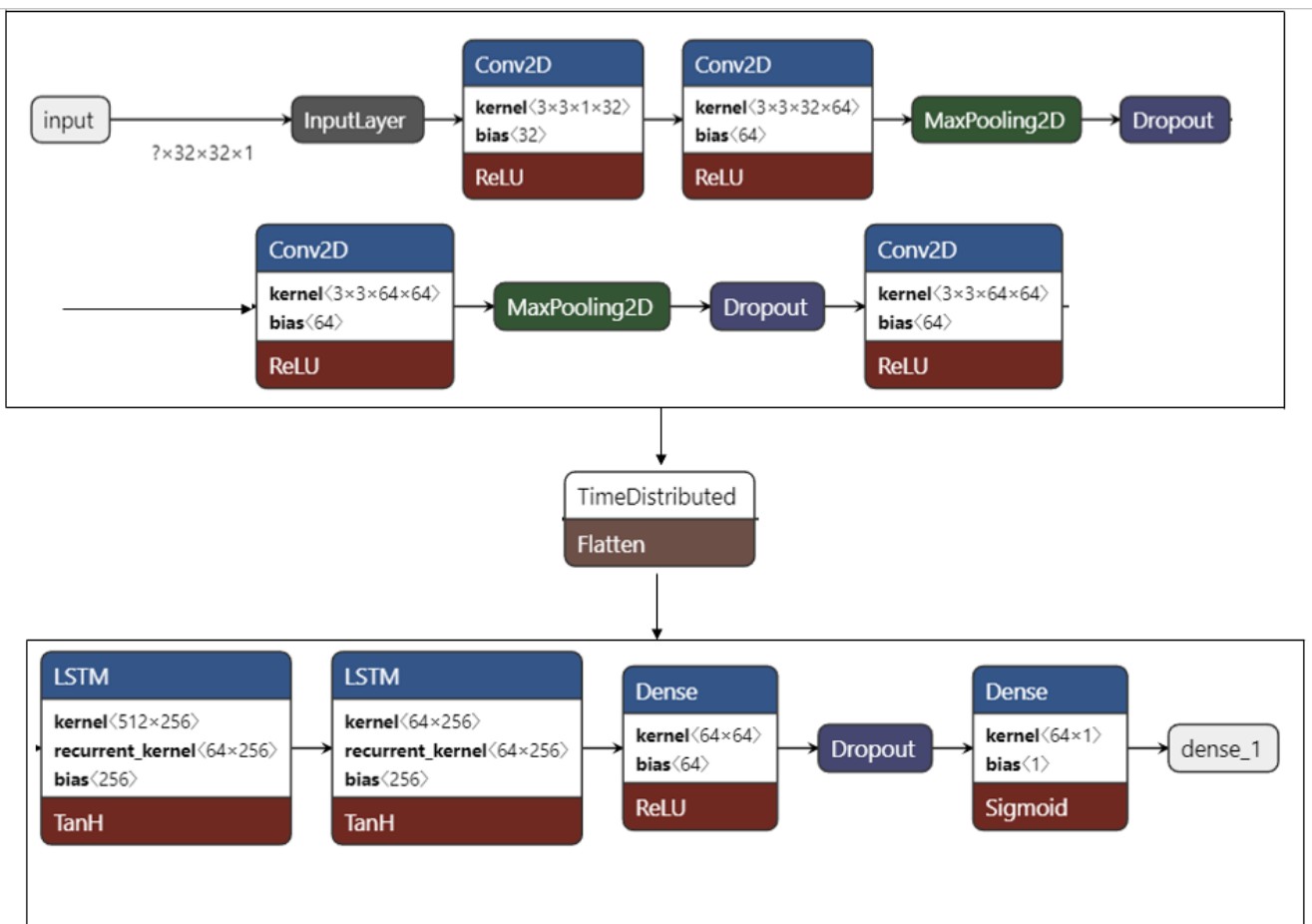

**Figure 12.** CNN–LSTM model architecture.

## 5. Experimental Results

### 5.1. MIMII Dataset

The MIMII dataset contained recordings of normal and abnormal operation sounds from industrial machines. Abnormal data were the recordings of abnormal sounds caused by contamination, leaks, rotation imbalance, or rail damage. Data types included fan, valve, pump, and slide rail. See Table 1. The dataset included approximately 5000–10,000 s of normal data and approximately 1000 s of abnormal data. For the MIMII dataset, four machines were recorded. Each machine had a model ID. In this study, only the data related to machines with a model ID of 00 was used. For the MIMII dataset, a circular array of eight microphones was used, and the distance of the microphone differed for each data type.

**Table 1.** Number of MIMII dataset recordings used.

| Type | Normal | Abnormal | Sum |
|---|---|---|---|
| Fan | 1011 | 407 | 1418 |
| Valve | 991 | 119 | 1110 |
| Pump | 1006 | 101 | 1107 |
| Slide rail | 1068 | 356 | 1424 |

Figure 13 visualizes the sound of the fan, slider, valve, and pump, beginning at the top left. Each of the eight microphones has a slightly distinct sound. Figure 14 displays the recording setup. The fan is closest to microphone number 5, the valve is closest to microphone number 1, the pump is closest to microphone number 3, and the sliding rail is closest to microphone number 7. Therefore, the results collected using the nearest microphone for each data point were used in this study. Furthermore, the experiment was performed in consideration of the poor environment by lowering the quality of the MIMII dataset. Thus, data quality was reduced from the original 16 K to 8 K and 4 K.

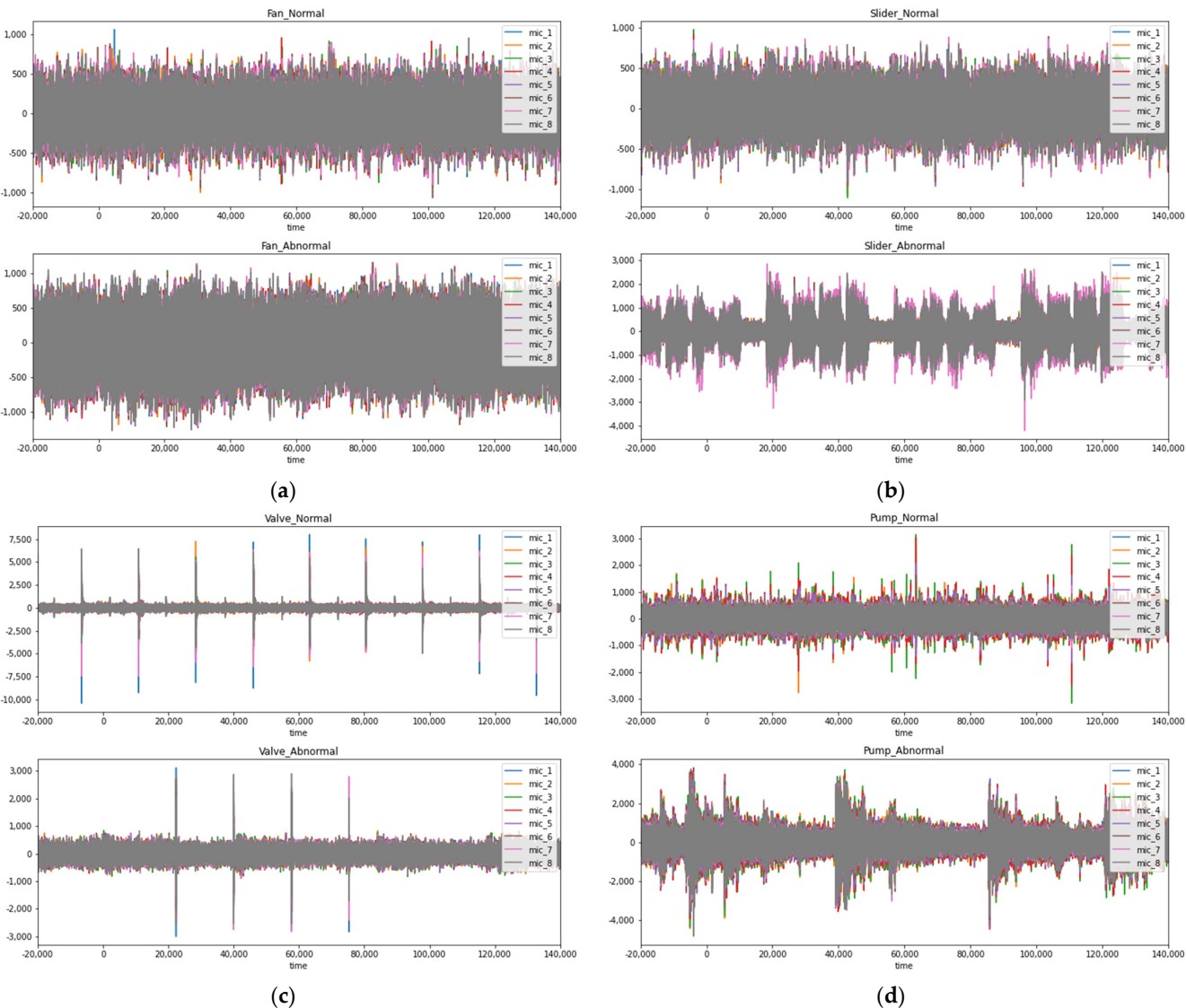

**Figure 13.** Visualizations of sound data for each device of the MIMII dataset: (**a**) normal and abnormal data for fan; (**b**) normal and abnormal data for slider; (**c**) normal and abnormal data for valve; (**d**) normal and abnormal data for pump.

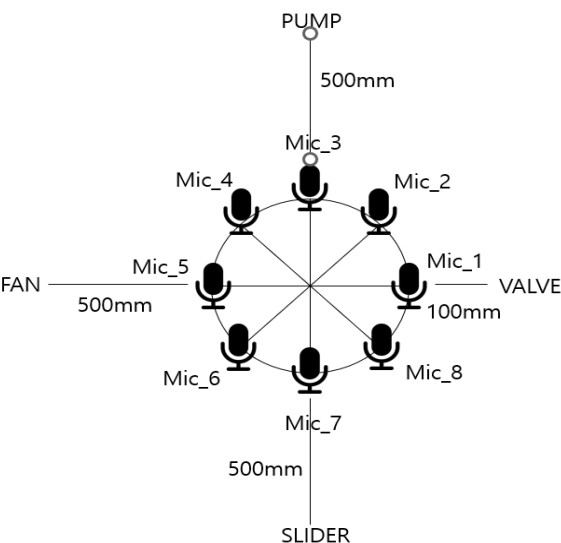

**Figure 14.** Data collecting environment for the MIMII dataset.

*5.2. Comparison between CNN and CNN–LSTM Models*

5.2.1. Comparison of Accuracy

We compared the items with the highest accuracy when using the same data. Figure 15 displays the outcomes of model training on valve and slide rail data from the MIMII dataset. The data were preprocessed using the STFT. With a high-quality sound and a SNR of 0 dB, both CNN and CNN–LSTM models performed well, achieving an accuracy of 1.0. Overall, the performance of the two models in this experiment was similar.

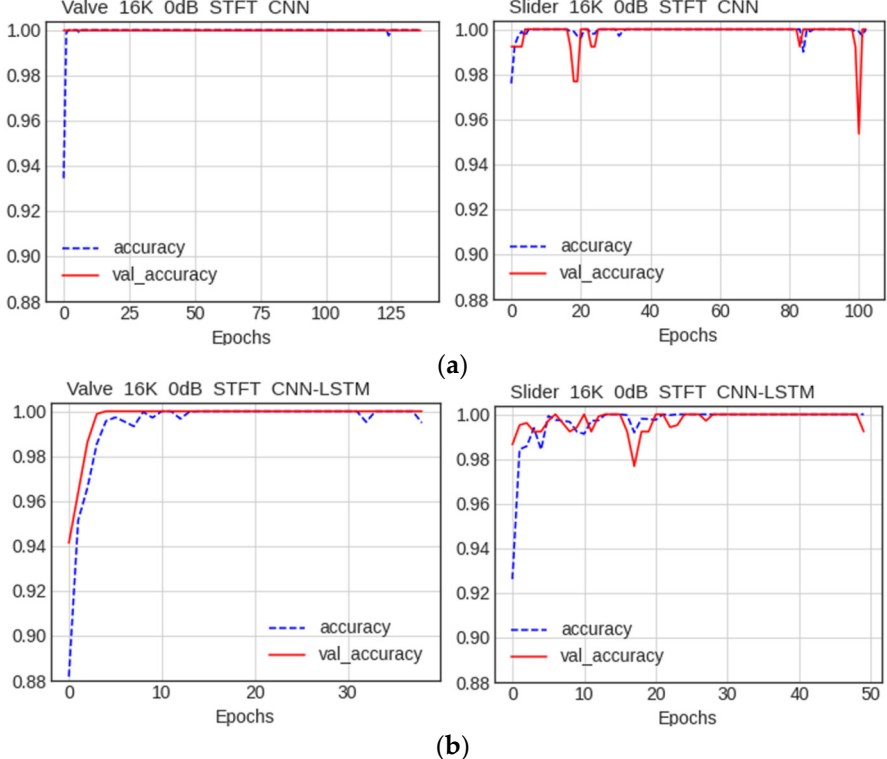

**Figure 15.** Accuracy of using the STFT data consisting of valves, and slide rail data (16 K): (**a**) results of training on the CNN model; and (**b**) results of training on the CNN–LSTM model.

### 5.2.2. Comparison of Performance by Condition

We compared the performances of the models to determine whether excellent performance was maintained even when new circumstances were introduced. For the first condition, performance was monitored when using different data preprocessing methods. For the second condition, we evaluated whether a high performance could be maintained even when the data quality was degraded. Finally, we evaluated whether high performance was maintained when introducing noise to simulate a real-world setting. The valve data were used for this comparison.

Comparison of Accuracy among Data Preprocessing Methods

The comparison of models, using the first condition, is displayed in Figure 16. The results were obtained by preprocessing the valve data using the DWT, the Hilbert transform, and the STFT, and training each model with the data. The CNN–LSTM model exhibited a superior performance on DWT data, compared to the CNN model. Both models performed well in the Hilbert transform and the STFT, and achieved the same accuracy of 1.0.

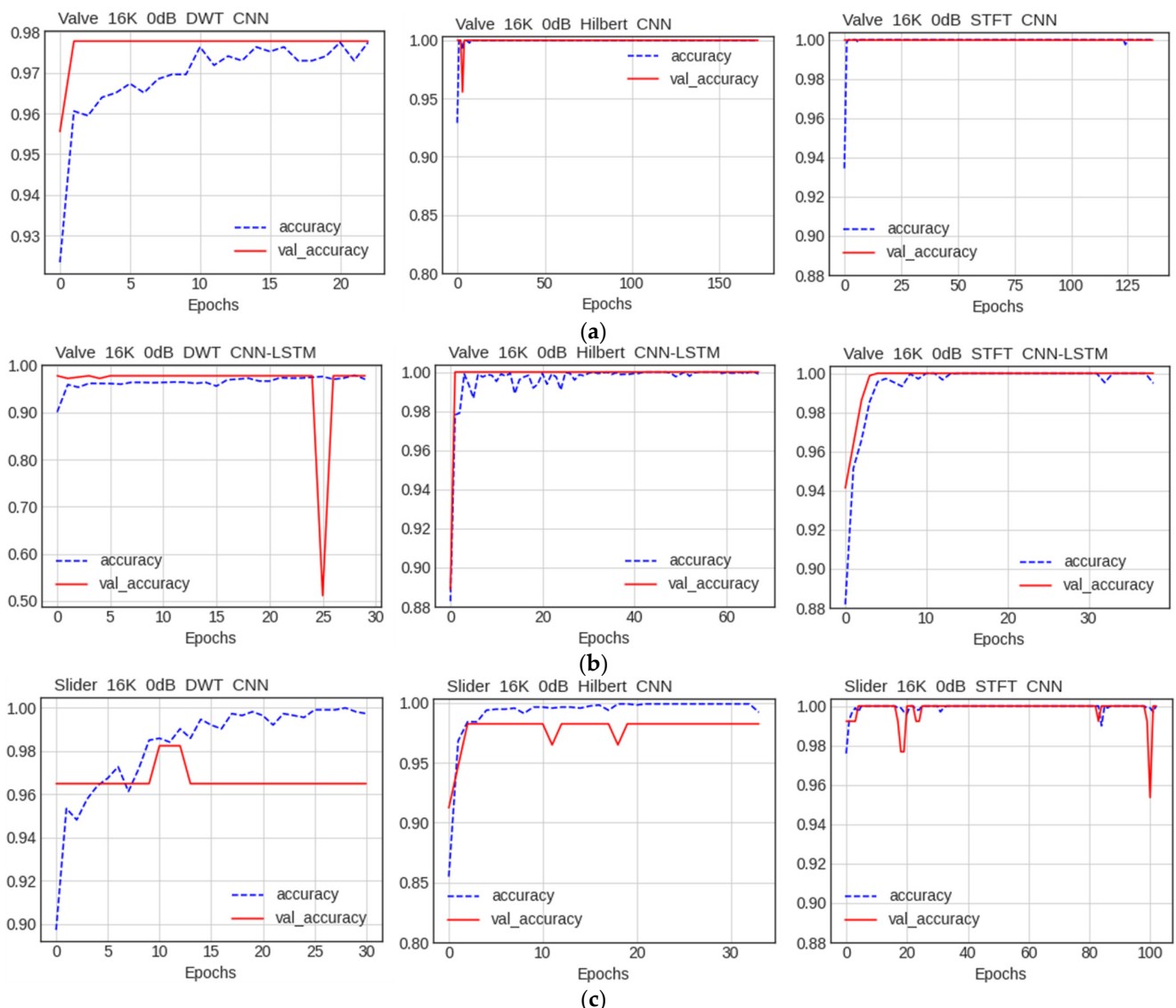

**Figure 16.** *Cont.*

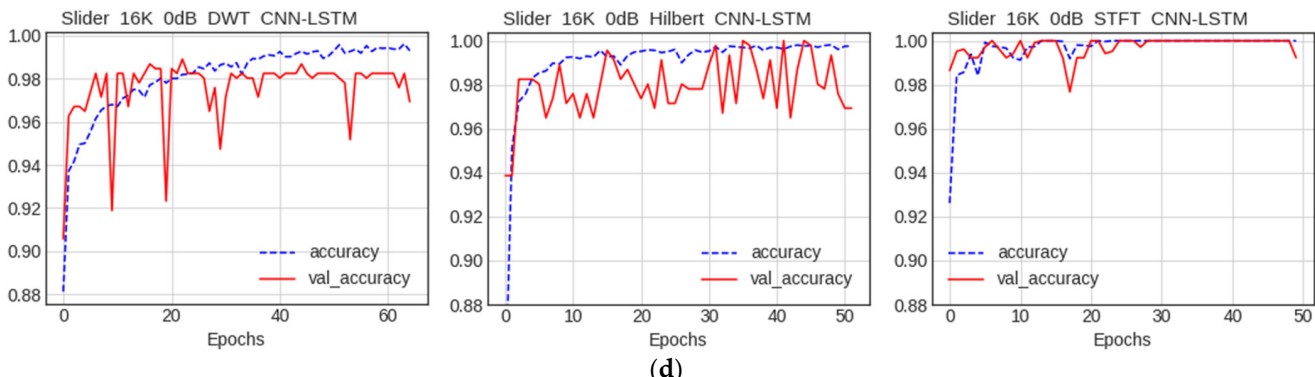

**Figure 16.** Accuracy of using data after preprocessing with DWT, Hilbert transform, and the STFT: (**a**) results of training valve data on CNN model; (**b**) results of training valve data on the CNN–LSTM model; (**c**) results of training slider data on CNN model; and (**d**) results of training slider data on the CNN–LSTM model.

Comparison of Accuracy by Data Quality

Figure 17 displays the results of changing the quality of the valve data to 4 K, 8 K, and 16 K, as well as the results of preprocessing using the Hilbert transform and training each model. From left to right, 16 K, 8 K, and 4 K data were used, and the accuracy of both models fell as the quality changed. Overall, the accuracy of the CNN–LSTM model continued to decline, but not significantly. Furthermore, because the same data were utilized, the accuracy of the CNN–LSTM model was excellent.

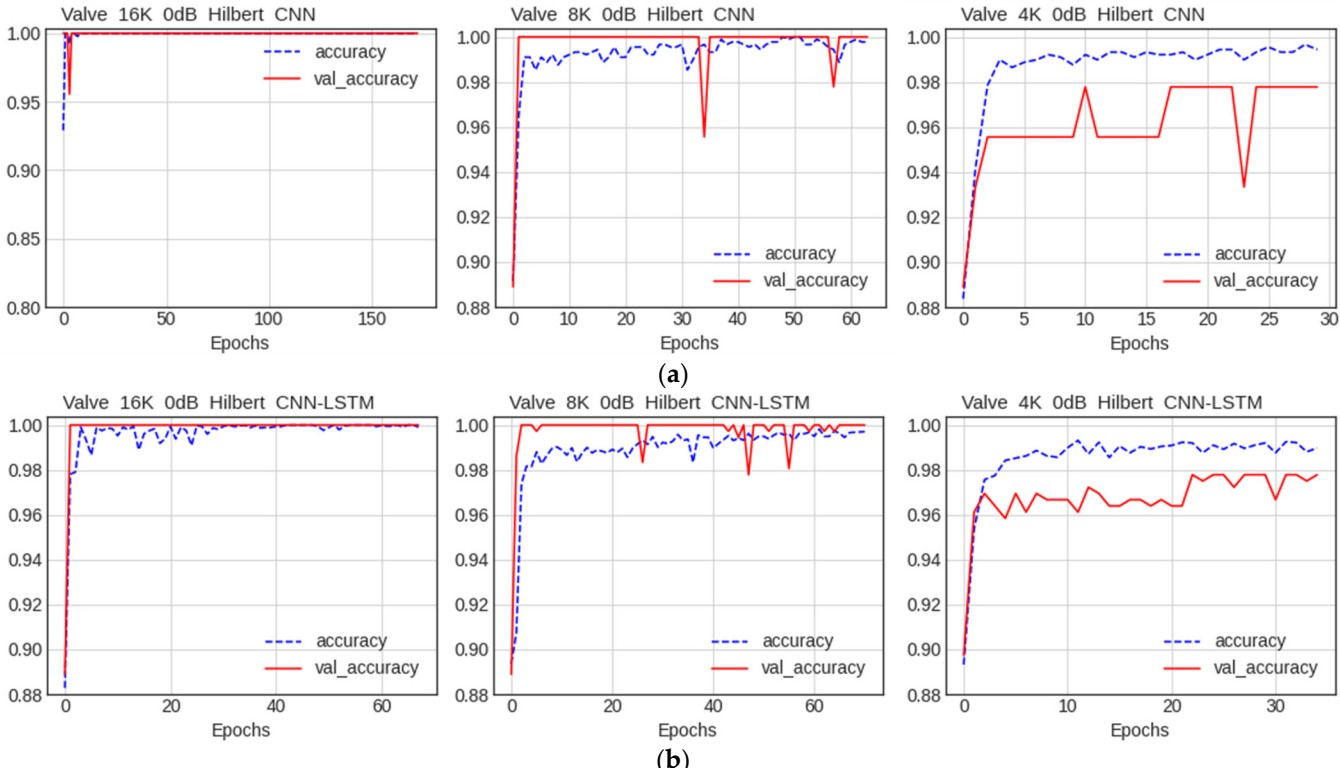

**Figure 17.** *Cont.*

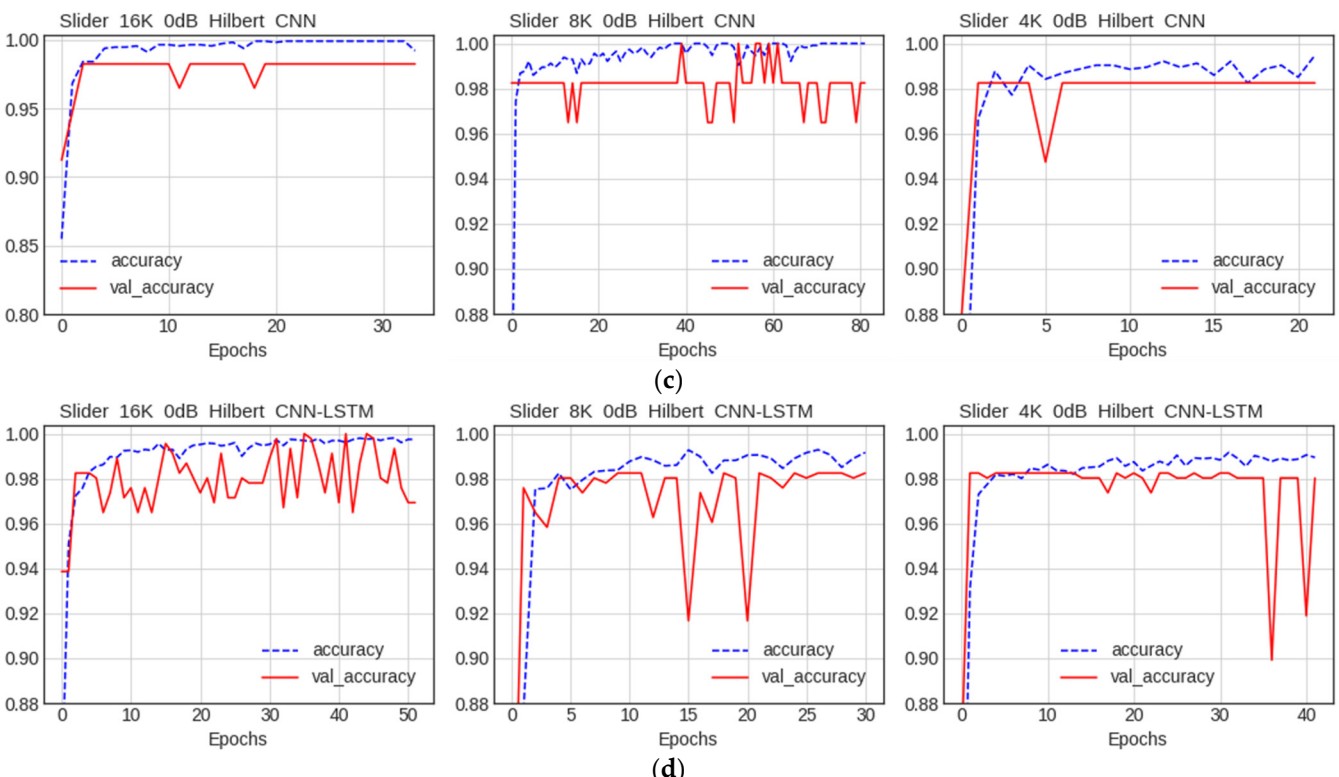

**Figure 17.** Accuracy of using data after preprocessing with Hilbert transform after changing the data quality to 4 K, 8 K, and 16 K: (**a**) result of valve data training on the CNN model; (**b**) result of valve data training on the CNN–LSTM model; (**c**) result of slider data training on the CNN model; and (**d**) result of slider data training on the CNN–LSTM model.

Next, the STFT on the same data, without using the Hilbert transform, was compared. Figure 18 displays the results of comparing the model; for this, the STFT was performed, using a different method to that used in the previous experiment, but using the same quality valve data as in the previous experiment. Unlike the Hilbert transform, the CNN model retained a high performance at 16 K, 8 K, and 4 K resolutions, with an accuracy of 1.0. The CNN–LSTM model performed poorly as the quality decreased. The two models performed differently for each preprocessing method.

Comparison of Accuracy by Noise

Finally, we compared the cases of adding noise (SNR: 0 dB, 6 dB, −6 dB). When the Hilbert transform and the STFT were performed in the previous experiment, the two models exhibited distinct performances. Therefore, both methods were compared using the valve data. The 8 K valve data were used because the performance was neither too good nor too bad. As displayed in Figure 19, the SNR was used by 0 dB, 6 dB, and −6 dB, in order from the upper left, and the accuracy of using the data after preprocessing with the Hilbert transform is displayed. When the SNR was 6 dB, the performance of both models increased slightly, and when the SNR was −6 dB, the performance decreased. We show that the CNN model has a lower performance than the CNN–LSTM model. When using Hilbert transformations, we show that the CNN model is vulnerable to noise.

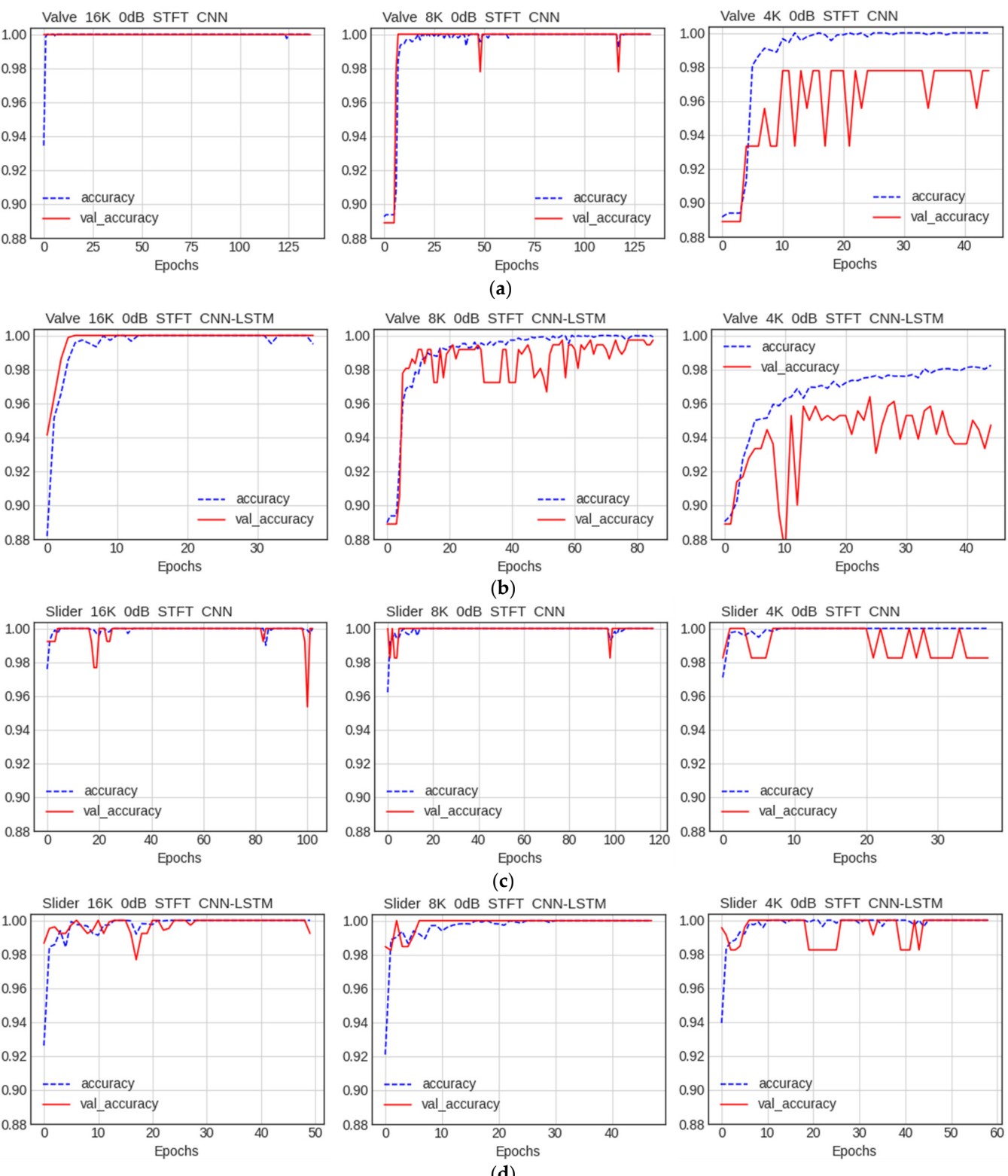

**Figure 18.** Accuracy of using data after preprocessing with the STFT after changing the data quality to 4 K, 8 K, and 16 K: (**a**) result of valve data training on the CNN model; (**b**) result of valve data training on the CNN–LSTM model; (**c**) result of slider data training on the CNN model; and (**d**) result of slider data training on the CNN–LSTM model.

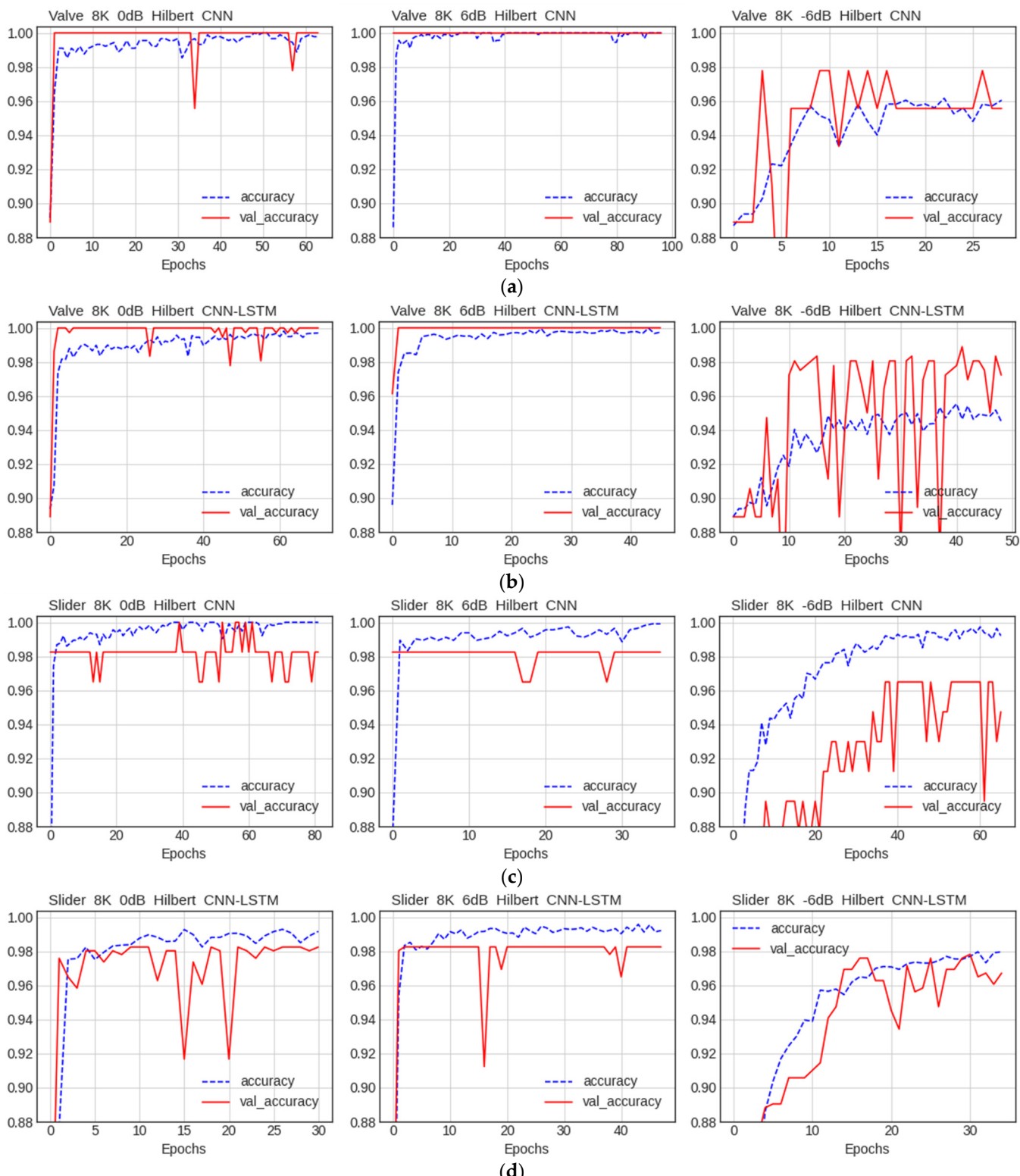

**Figure 19.** Accuracy of using data after preprocessing with the Hilbert transform (8 K, SNR: 0 dB, 6 dB, −6 dB): (**a**) results of valve data training on the CNN model; (**b**) results of valve data training on the CNN–LSTM model; (**c**) results of slider data training on the CNN model; and (**d**) results of slider data training on the CNN–LSTM model.

Next, the STFT was performed on the same data and the results were compared. As in the last experiment, the SNR of 0 dB, 6 dB, and −6 dB were used. As illustrated in Figure 20,

the accuracy did not decrease when the SNR was 6 dB, but the accuracy of the two models decreased slightly when the SNR was −6 dB. Both models showed similar accuracy in noisy situations.

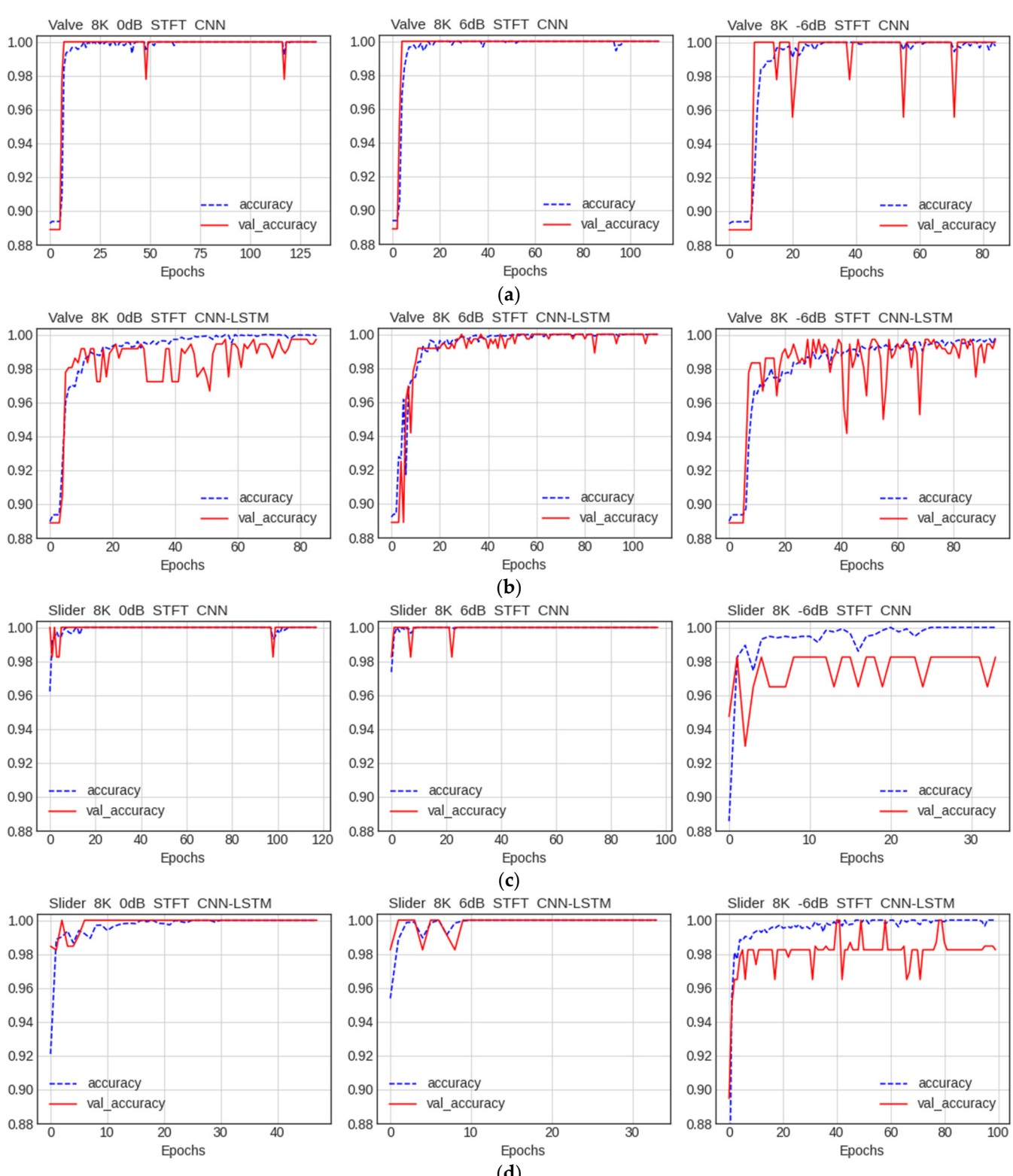

**Figure 20.** Accuracy of using data after preprocessing with the STFT transform (8 K, SNR: 0 dB, 6 dB, −6 dB): (**a**) results of valve data training on the CNN model; (**b**) results of valve data training on the CNN–LSTM model; (**c**) results of slider data training on the CNN model; and (**d**) results of slider data training on the CNN–LSTM model.

In all three tests, the CNN–LSTM model was similar to the CNN model in terms of accuracy by preprocessing, accuracy by data quality, and accuracy by noise. However, when the two models were used in a real factory, the accuracy levels were not significantly different, which indicated that the difference could be attributed to the lightweight characteristics of the model. Given the small number of parameters of the CNN–LSTM model compared to the CNN model, the CNN–LSTM model is easy to use in real-world applications. This is because it can operate even in a system with limited memory and bandwidth.

### 5.3. Model Evaluation

Based on the preceding comparison section, we compared CNN and CNN–LSTM models using the Hilbert transform and the STFT method. The precision, recall, and F1 score were used to evaluate the models. Table 2 details the results of model training on the Hilbert transform of the valve data. Table 3 presents the outcomes of model training on valve data after applying the STFT. Looking at the results of the table, the performance indicator results slightly decrease when the sound is low quality. It can also be seen that the performance varies depending on the noise. As a result of comparing the two models in this way, there was a performance difference of up to approximately 0.5%, based on F1 score, and there was a performance overall.

**Table 2.** Result of training on the CNN and CNN–LSTM models by performing Hilbert transform for valve data.

| Quality | Model | SNR | Accuracy | Precision | Recall | F1 Score |
|---------|-------|-----|----------|-----------|--------|----------|
| **16 K** | CNN | 0 dB | 1.0 | 1.0 | 1.0 | 1.0 |
| | | 6 dB | 0.9943 | 0.9937 | 1.0 | 0.9968 |
| | | −6 dB | 1.0 | 1.0 | 1.0 | 1.0 |
| | CNN–LSTM | 0 dB | 1.0 | 1.0 | 1.0 | 1.0 |
| | | 6 dB | 0.9950 | 0.9936 | 0.9936 | 0.9936 |
| | | −6 dB | 0.9992 | 1.0 | 0.9936 | 0.9968 |
| **8 K** | CNN | 0 dB | 0.9887 | 1.0 | 0.9874 | 0.9936 |
| | | 6 dB | 1.0 | 1.0 | 1.0 | 1.0 |
| | | −6 dB | 0.9491 | 0.9629 | 0.9811 | 0.9719 |
| | CNN–LSTM | 0 dB | 0.9992 | 1.0 | 1.0 | 1.0 |
| | | 6 dB | 1.0 | 1.0 | 1.0 | 1.0 |
| | | −6 dB | 0.9272 | 0.9239 | 0.9937 | 0.9575 |
| **4 K** | CNN | 0 dB | 0.9887 | 1.0 | 0.9937 | 0.9968 |
| | | 6 dB | 1.0 | 1.0 | 1.0 | 1.0 |
| | | −6 dB | 0.9039 | 0.9277 | 0.9685 | 0.9476 |
| | CNN–LSTM | 0 dB | 0.9936 | 1.0 | 0.9937 | 0.9968 |
| | | 6 dB | 1.0 | 1.0 | 1.0 | 1.0 |
| | | −6 dB | 0.8997 | 0.8983 | 1.0 | 0.9464 |

**Table 3.** Result of training on the CNN and CNN–LSTM models by performing the STFT for valve data.

| Quality | Model | SNR | Accuracy | Precision | Recall | F1 Score |
|---------|-------|-----|----------|-----------|--------|----------|
| **16 K** | CNN | 0 dB | 1.0 | 1.0 | 1.0 | 1.0 |
| | | 6 dB | 1.0 | 1.0 | 1.0 | 1.0 |
| | | −6 dB | 1.0 | 1.0 | 1.0 | 1.0 |
| | CNN–LSTM | 0 dB | 1.0 | 1.0 | 1.0 | 1.0 |
| | | 6 dB | 0.9983 | 0.9952 | 1.0 | 0.9976 |
| | | −6 dB | 0.9983 | 0.9952 | 1.0 | 0.9976 |
| **8 K** | CNN | 0 dB | 1.0 | 0.9937 | 1.0 | 0.9968 |
| | | 6 dB | 1.0 | 1.0 | 1.0 | 1.0 |
| | | −6 dB | 0.9717 | 0.9753 | 0.9937 | 0.9844 |
| | CNN–LSTM | 0 dB | 0.9908 | 1.0 | 1.0 | 1.0 |
| | | 6 dB | 0.9992 | 1.0 | 0.9936 | 0.9968 |
| | | −6 dB | 0.9632 | 0.9277 | 0.9685 | 0.9476 |
| **4 K** | CNN | 0 dB | 1.0 | 0.9298 | 1.0 | 0.9636 |
| | | 6 dB | 1.0 | 1.0 | 1.0 | 1.0 |
| | | −6 dB | 0.9661 | 0.9751 | 0.9874 | 0.9812 |
| | CNN–LSTM | 0 dB | 0.9654 | 1.0 | 1.0 | 1.0 |
| | | 6 dB | 0.9985 | 1.0 | 1.0 | 1.0 |
| | | −6 dB | 0.9435 | 0.8983 | 1.0 | 0.9464 |

Table 4 shows the results of model training on Hilbert transformations of slider data, and Table 5 shows the results of model training on slider data after the STFT application. Table results show that the STFT method outperforms the Hilbert method, although it is similar to the valve data.

**Table 4.** Result of training on the CNN and CNN–LSTM models by performing Hilbert transform for slider data.

| Quality | Model | SNR | Accuracy | Precision | Recall | F1 Score |
|---------|-------|-----|----------|-----------|--------|----------|
| **16 K** | CNN | 0 dB | 0.9912 | 0.9941 | 0.9941 | 0.9941 |
| | | 6 dB | 0.9956 | 1.0 | 0.9941 | 0.9970 |
| | | −6 dB | 0.9824 | 1.0 | 0.9766 | 0.9881 |
| | CNN–LSTM | 0 dB | 0.9923 | 1.0 | 0.9883 | 0.9941 |
| | | 6 dB | 0.9945 | 1.0 | 0.9941 | 0.9970 |
| | | −6 dB | 0.9868 | 0.9884 | 1.0 | 0.9941 |
| **8 K** | CNN | 0 dB | 0.9868 | 0.9883 | 0.9941 | 0.9912 |
| | | 6 dB | 0.9955 | 0.9941 | 1.0 | 0.9970 |
| | | −6 dB | 0.9691 | 0.9657 | 0.9941 | 0.9797 |
| | CNN–LSTM | 0 dB | 0.9780 | 0.9883 | 0.9824 | 0.9853 |
| | | 6 dB | 0.9906 | 0.9883 | 1.0 | 0.9941 |
| | | −6 dB | 0.9768 | 0.9604 | 1.0 | 0.9798 |
| **4 K** | CNN | 0 dB | 0.9911 | 0.9883 | 1.0 | 0.9941 |
| | | 6 dB | 0.9911 | 0.9941 | 0.9941 | 0.9941 |
| | | −6 dB | 0.9647 | 0.9602 | 0.9941 | 0.9768 |
| | CNN–LSTM | 0 dB | 0.9895 | 0.9882 | 0.9882 | 0.9882 |
| | | 6 dB | 0.9911 | 0.9883 | 1.0 | 0.9941 |
| | | −6 dB | 0.9576 | 0.9491 | 0.9882 | 0.9682 |

**Table 5.** Result of training on the CNN and CNN–LSTM models by performing the STFT for slider data.

| Quality | Model | SNR | Accuracy | Precision | Recall | F1 Score |
|---------|-------|-----|----------|-----------|--------|----------|
| **16 K** | CNN | 0 dB | 1.0 | 1.0 | 1.0 | 1.0 |
| | | 6 dB | 1.0 | 1.0 | 1.0 | 1.0 |
| | | −6 dB | 0.9932 | 0.9911 | 1.0 | 0.9955 |
| | CNN–LSTM | 0 dB | 1.0 | 1.0 | 1.0 | 1.0 |
| | | 6 dB | 1.0 | 1.0 | 1.0 | 1.0 |
| | | −6 dB | 0.9932 | 0.9911 | 1.0 | 0.9955 |
| **8 K** | CNN | 0 dB | 1.0 | 1.0 | 1.0 | 1.0 |
| | | 6 dB | 1.0 | 1.0 | 1.0 | 1.0 |
| | | −6 dB | 0.9823 | 0.9940 | 0.9823 | 0.9881 |
| | CNN–LSTM | 0 dB | 1.0 | 1.0 | 1.0 | 1.0 |
| | | 6 dB | 1.0 | 1.0 | 1.0 | 1.0 |
| | | −6 dB | 0.9911 | 0.9941 | 0.9941 | 0.9941 |
| **4 K** | CNN | 0 dB | 1.0 | 1.0 | 1.0 | 1.0 |
| | | 6 dB | 1.0 | 1.0 | 1.0 | 1.0 |
| | | −6 dB | 0.9955 | 1.0 | 0.9941 | 0.9970 |
| | CNN–LSTM | 0 dB | 1.0 | 1.0 | 1.0 | 1.0 |
| | | 6 dB | 1.0 | 1.0 | 1.0 | 1.0 |
| | | −6 dB | 0.9955 | 1.0 | 0.9941 | 0.9970 |

However, when the number of parameters of the two models were considered, the lightweight CNN–LSTM model was preferable, because it used far fewer parameters than the CNN model. This lightweight property is crucial in deciding whether the model can be run on AIoT equipment or accessed via an embedded system.

## 6. Discussion and Comparison with Similar Works

We propose a fault-diagnostic deep learning model that is operational within a real AIoT platform. There a necessity for a model capable of edge computing in the numerous manufacturing sites that exist in the industry. However, previous studies have only tried to improve diagnostic classification performance [8–11]. We have conducted a study that significantly reduces the size of the model, with minimal performance degradation, to be applicable in the real industry. This is because if the size of the model is large, it is impossible to operate on edge computing equipment with a limited memory capacity.

Table 6 shows a comparison of our model with the model proposed in previous studies. The WMV model [9] used ensemble techniques based on the pre-learning model EfficientNet. However, since this model has at least 96.5 million parameters, it is very large and difficult to use in real industry. Our CNN model uses a $32 \times 32$ small input size and has approximately 4.2 million parameters. The CNN model uses a very small number of parameters, less than 4.4% compared to the WMV model. The SCRLSTM model [8] was constructed by applying the Mel-spectrogram method to receive input as an image and combining the LSTM model. This model has a small number of parameters, of 1,045,330. Our proposed CNN–LSTM model uses a smaller input size and significantly reduces the number of parameters to 277,633. In other words, it made it possible to operate on an AIoT platform that uses very small memory.

**Table 6.** Comparison between proposed and existing models.

| Model | WMV [9] | CNN (Ours) | SCRLSTM [8] | CNN–LSTM (Ours) |
|---|---|---|---|---|
| Feature Extraction | MFCC | DWT, Hilbert, STFT | Mel-spectrogram | DWT, Hilbert, STFT |
| Architecture | EfficientNet-B0 (4 millions) EfficientNet-B5 (28.5 millions) EfficientNet-B7 (64 millions) | Conv2D (32, 32, 32) Conv2D (32, 32, 64) MaxPooling2D (16, 16, 64) Dropout (16, 16, 64) Flatten (16,384) Dense (256) Dropout (256) Dense (1) | Conv2D (360, 144, 10) MaxPooling2D (180, 72, 10) Conv1D (180, 72, 54) TimeDistributed (180, 3888) LSTM (180, 54) LSTM (180, 54) LSTM (180, 108) LSTM (180, 108) LSTM (108, 2) | Conv2D (32, 32, 32) Conv2D (32, 32, 64) MaxPooling2D (16, 16, 64) Dropout (16, 16, 64) Conv2D (16, 16, 64) MaxPooling2D (8, 8, 64) Dropout (8, 8, 64) Conv2D (8, 8, 64) TimeDistributed (8, 512) LSTM (8, 64) LSTM (8, 64) Dense (8, 64) Dropout (8, 64) Dense (8, 1) |
| Number of Parameters | 96,500,000~ | 4,213,633 | 1,045,330 | 277,633 |

## 7. Conclusions and Future Work

In this study, we propose a deep learning model for fault diagnosis that can be used in the real industry. To the best of our knowledge, there were no studies considering lightening in deep learning model-based fault diagnosis, and no experiments comparing DWT, Hilbert, and the STFT as feature extraction methods. In addition, there have been no studies of fault diagnosis models that behave robustly in sound quality degradation and additional noise environments. Our proposed lightweight model showed that the performance degradation is very small but the number of parameters is greatly reduced, enabling practical use. In addition, it showed good performance even in low-quality sound and noisy situations. The results of this study are thought to be meaningful when applied to AIoT platforms with limited memory space and network bandwidth in the real industry.

In the future, we will mount the proposed lightweight fault diagnosis deep learning model on the AIoT platform. And we will further learn various datasets to develop our model. In the future, we plan to study domain adaptation and self-supervised learning that can operate in other equipment and environments.

**Author Contributions:** Conceptualization, S.L.; methodology, S.L.; software, J.S.; validation, S.L.; formal analysis, J.S.; investigation, J.S.; resources, J.S.; data curation, S.L.; writing—original draft preparation, J.S.; writing—review and editing, S.L.; visualization, J.S.; supervision, S.L.; project administration, S.L.; funding acquisition, S.L. All authors have read and agreed to the published version of the manuscript.

**Funding:** This study was supported by the "Regional Innovation Strategy (RIS)" through the National Research Foundation of Korea (NRF), funded by the Ministry of Education (MOE) 2021RIS-001 (1345341783).

**Institutional Review Board Statement:** Not applicable.

**Informed Consent Statement:** Not applicable.

**Data Availability Statement:** The MIMII dataset is openly available at: https://zenodo.org/record/3384388 (accessed on 7 January 2023).

**Conflicts of Interest:** The authors declare no conflict of interest.

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
