# Peer review of "Robust and Lightweight Deep Learning Model for Industrial Fault Diagnosis in Low-Quality and Noisy Data"

_electronics, doi:10.3390/electronics12020409_

Round 1

Reviewer 1 Report

The author proposed a variation of CNN to claim it as light weight to solve the problem of industrial machine malfunctioning based on the sound produced by machine in the presence of noise. Its a useful work, however there are certain observations based on which this work needs to be updated before further consideration.

1. What is this 0.005? mentioned in abstract. What sort of difference it is? It must be some parameter or have unit? Mention it to clarify.

2. Authors mentioned that CNN is heavy and computationally complex as it use hundred of thousands of features while the modified CNN, i.e. CNN-LSTM use only fraction of features compared to CNN. That naturally makes the CNN-LSTM a lighter version and computationally fast. But the question here is that, authors simply compared CNN with CNN-LSTM (for the addressed problem), have anyone tried CNN for this problem? If so what kind of results are achieved and what was the computational complexity.

3. Authors Must compare their produced results with state of the art algorithms that are used to solve the same problem using same dataset as authors mentioned. Only then this will be a fair comparison and shall show if the proposed model is better or not. 

4. Accuracy and computational complexity, both measures should be used to show results to establish the superiority of the proposed methods as compared to state of the art.

Author Response

Thank you for your professional review. We wrote and uploaded a word file to respond to your comment sincerely.

Reviewer 2 Report

The analysis and investigations carried out in this manuscript cover an interesting topic on “Robust and Lightweight Deep Learning Model for Industrial 2 Fault Diagnosis in Low-quality and Noisy Data”. However, this manuscript needs some improvements, as described in the following sections, before it can be accepted for publication.

The following points may be taken into account for the improvement of the manuscript:

  1. Abstract lack of information. The background study, problem statement, points out research gaps, aims & objectives, a summary of methods, and novelty of the research study are not clearly presented.
  2. Abstract. MIMII, CNN-LSTM, full name, and following by abbreviation.
  3. Introduction part is lengthy. It should be informative and crisp.
  4. Introduction. Some of the sentences need to support with references.
  5. In the Introduction section, there are few lines without any literature support, such as given below:
  6. Conclusions need to be point-wise. At least, from each result and discussion section, one outcome should be presented in the conclusions section.    
  7. No Validation for the results. No discussions.
  8. not found any scientific reasons for drawing the conclusions.
  9. No comparison with previous works.
  10. Conclusion should include how effectively the methodology & process parameters were selected, and validated and can be used for future application.
  11. Few more references need to be included in body text.

12.  Pls Mention the X and Y axis scales clearly and make the Figure larger to make it readable.

Author Response

(The authors gave the same response as above.)

Reviewer 3 Report

This paper is strong, and has mathematical real-world applications; I see many positive aspects in this work and would like to see it published. In fact, this paper will be of value and interest to as a significant portion of potential readers of the journal. In my opinion, this paper can be further improved in the following aspects:

1. The contributions should be more clearly explained with more details on how to improve the existing results, especially in the references the authors cited.

2. Some parts of mathematic derivations are not given in details. I suggest the authors a carefully checking and give all the necessary manipulations for the derived formulas.

3. More detailed review of the literature is expected in separate section. Specially, it is required that the previous solutions to this problem be addressed. Then, the advantages (and disadvantages?) of the proposed methods and should be discussed.

4. Some latest references about intelligence data analysis and fault diagnosis should be added to give readers an up-to-date picture. In this sense, the following papers can be referred: Domain-adaptive intelligence for fault diagnosis based on deep transfer learning from scientific test rigs to industrial applications,NCAA; Sparsity-based signal extraction using dual Q-factors for gearbox fault detection.

5. The authors should clarify if the proposed method requires data processing or not. More detail information should be added to make clear explanations. 

6. Some future directions can be discussed in the conclusion part.

I recommend the publication of this paper after the authors address the above concerns.

Author Response

(The authors gave the same response as above.)

Round 2

Reviewer 1 Report

Its improved and good to go.